# Technological Quality of Sugarcane Inoculated with Plant-Growth-Promoting Bacteria and Residual Effect of Phosphorus Rates

**DOI:** 10.3390/plants12142699

**Published:** 2023-07-20

**Authors:** Guilherme Carlos Fernandes, Poliana Aparecida Leonel Rosa, Arshad Jalal, Carlos Eduardo da Silva Oliveira, Fernando Shintate Galindo, Ronaldo da Silva Viana, Pedro Henrique Gomes De Carvalho, Edson Cabral da Silva, Thiago Assis Rodrigues Nogueira, Abdulaziz A. Al-Askar, Amr H. Hashem, Hamada AbdElgawad, Marcelo Carvalho Minhoto Teixeira Filho

**Affiliations:** 1Department of Plant Health, College of Engineering, São Paulo State University (UNESP), Rural Engineering and Soils, Ilha Solteira 15385-000, SP, Brazil; guilherme.carlos-fernandes@unesp.br (G.C.F.); polirosa1@hotmail.com (P.A.L.R.); arshad.jalal@unesp.br (A.J.); ces.oliveira@unesp.br (C.E.d.S.O.); pedro.goomes04@gmail.com (P.H.G.D.C.); edsoncabralsilva@gmail.com (E.C.d.S.); tar.nogueira@unesp.br (T.A.R.N.); 2Department of Plant Production, Faculty of Agricultural and Technological Sciences, São Paulo State University (UNESP), Dracena 17900-000, SP, Brazil; fernando.galindo@unesp.br (F.S.G.); ronaldo.viana@unesp.br (R.d.S.V.); 3Department of Agricultural Sciences, School of Agricultural and Veterinarian Sciences, São Paulo State University (UNESP), Via do Prof Access Paulo Donato Castellane, s/n, Jabotcabal 14884-900, SP, Brazil; 4Department of Botany and Microbiology, Faculty of Science, King Saud University, Riyadh 11451, Saudi Arabia; aalaskara@ksu.edu.sa; 5Botany and Microbiology Department, Faculty of Science, Al-Azhar University, Cairo 11884, Egypt; amr.hosny86@azhar.edu.eg; 6Integrated Molecular Plant Physiology Research, Department of Biology, University of Antwerp, 2020 Antwerp, Belgium; hamada.abdelgawad@uantwerpen.be

**Keywords:** PGPBs, ratoon cane, residual phosphorus, *Saccharum* spp., sugarcane quality

## Abstract

Phosphate fertilization in highly weathered soils has been a major challenge for sugarcane production. The objective of this work was to evaluate the foliar levels of phosphorus (P) and nitrogen (N) and the technological quality and productivity of second ratoon cane as a function of inoculation with plant-growth-promoting bacteria (PGPBs) together with the residual effect of phosphate fertilization. The experiment was carried out at the research and extension farm of Ilha Solteira, state of São Paulo, Brazil. The experiment was designed in a randomized block with three replications in a 5 × 8 factorial scheme. The treatments consisted of five residual doses of phosphorus (0, 45, 90, 135 and 180 kg ha^−1^ of P_2_O_5_, 46% P) applied at planting from the source of triple superphosphate and eight inoculations from three species of PGPB (*Azospirillum brasilense*, *Bacillus subtilis* and *Pseudomonas fluorescens*), applied in single or co-inoculation at the base of stems of sugarcane variety RB92579. Inoculation with PGPBs influenced leaf N concentration, while inoculations with *Pseudomonas fluorescens* and combinations of bacteria together with the highest doses exerted a positive effect on leaf P concentration. Co-inoculation with *A. brasilense* + *Pseudomonas fluorescens* associated with a residual dose of 135 kg ha^−1^ of P_2_O_5_ increased stem productivity by 42%. Thus, it was concluded that inoculations with *Pseudomonas fluorescens* and their combinations are beneficial for the sugarcane crop, reducing phosphate fertilization and increasing productivity.

## 1. Introduction

Brazil is the world’s largest producer of sugarcane (*Saccharum officinarum*), with a total production of 654.5 million tons in the 2020/2021 harvest. The state of São Paulo is ranked as the largest national producer of sugarcane with a total of 354.3 million tons and an average of 79.719 kg ha^−1^ [1]. Sugarcane (*Saccharum* sp.) is a semi-perennial crop which provides cuts and production cycles for an average of six years [2]. The first cut generated from seedlings is called cane plants and the other cuts are called ratoon cane. In addition, sugarcane is a viable alternative source of biofuel and energy generation given the increasing worldwide concern about climate change and growing demand for sustainable production. Sugarcane is one of the main agricultural crops to produce sugar and ethanol [1], in addition to other products such as liquor, molasses, brown sugar and by-products such as bagasse that are used for the co-generation of energy [3]. 

Sugarcane, like many other crops, has phosphorus (P) as the nutrient that most limits its productivity. In addition, it is a great challenge to ensure that P is in its labile form and available in the soil solution, which directly interferes with crop production [4]. The application of adequate doses of phosphate fertilizers is of great importance for vegetative development, generating more productive stumps and increasing the durability of the sugarcane crop [5]. The low availability of P during several years of sugarcane cultivation is due to several factors such as source, doses, low natural availability, adsorption to clay particles and precipitation with iron (Fe) and aluminum (Al) oxides [6].

Phosphorus plays an important role in sugarcane metabolism, being specifically involved in protein formation, cell division, photosynthesis, adenosine triphosphate (ATP) synthesis, sugar breakdown, respiration and sucrose formation as well as favoring rooting, tillering, final yield, sugar production and absorption of other nutrients [7]. In addition to benefits in the field, P fertilization is greatly important for the quality of sugarcane, influencing sucrose percentage and juice purity [8]. The quality of raw material is defined by a set of characteristics that consider the demands of industry at the time of processing, especially sucrose and industrial fiber contents [9]. Consequently, there is a need to apply high P doses, but the efficiency of P fertilization is still very low. In this way, new techniques are needed to adapt to increased P fertilization efficiency and its residual effect along sugarcane cuts, especially in the regions of less fertility and highly weathered soils with limited P availability, to reduce high doses fertilization and inputs costs.

Sustainable management efforts aim to maintain the ecosystem and biodiversity, seeking possibilities to reduce possible negative environmental impacts resulting from the continuous use of fertilizers. Therefore, a more sustainable way is the use of plant-growth-promoting bacteria (PGPBs) that can contribute to the growth of cultures through the synthesis of phytohormones and secondary metabolites, better absorption and assimilation of water and nutrients, solubilization of nutrients such as P, zinc (Zn) and potassium (K), promotion of biological nitrogen fixation and induction of tolerance to abiotic and biotic stresses [10,11,12,13,14]. 

Some microorganisms can transform insoluble phosphorus into ions that can be absorbed by plants through the processes of solubilization and mineralization of inorganic phosphorus (Pi) and organic phosphorus (Po) present in the soil, respectively. The BPCPs that could solubilize Pi carry out this process through the secretion of low-molecular-weight organic acids, such as glycolic, lactic, citric, and others [15]. The process of mineralization of Po, in turn, occurs after the decomposition of organic matter present in the soil, of which the final product will be a series of organic molecules containing Po, which are then dephosphorylated by bacteria that produce phosphatases [16]. Bacteria of the genera *Arthrobacter*, *Bacillus*, *Pseudomonas* and *Rhizobium*, among others, have demonstrated the ability to solubilize phosphates [17], thus having potential for the better use of phosphorus present in the solid phase of the soil, constituting viable alternatives for the inoculation of crops. However, it is noteworthy that the genera *Bacillus*, *Pseudomonas* and *Rhizobium* are considered the most efficient solubilizers of inorganic phosphates [15]. 

Among the combinations of PGPBs, co-inoculation of *Azospirillum brasilense* and *Bacillus subtilis* has shown positive results in nutrient cycling and soil fertility improvement, thus contributing to better sugarcane root development [18]. The *Bacillus* and *Pseudomonas* genera are the most efficient phosphate solubilizing inoculants [19] due to their role in the production of organic acids and solubilization of insoluble inorganic phosphate compounds such as tricalcium phosphate, dicalcium phosphate, hydroxyapatite and rock phosphate [17,20], and their phosphatase activity to mineralize organic phosphorus [15], thus improving the residual effect of phosphorus fertilization. 

It is believed that the interactions between the species of PGPBs of the genera *Azospirillum*, *Bacillus* and *Pseudomonas* have the capacity to increase the efficiency of fertilization with residual P in rat cane, with positive effects on the productivity and technological quality of the sugarcane crop [21]. In this context, the present objective was to evaluate the effect of inoculation with three species of plant-growth-promoting bacteria, applied alone or in combination with residual doses of P_2_O_5_ (applied at the time of planting) on foliar nitrogen (N) and phosphorus (P) concentrations, yield of stalks, sugar and technological quality of cane second ratoon in tropical soil with low P content.

## 2. Results

### 2.1. Leaf Concentrations of N and P as a Function of Inoculations and P_2_O_5_ Doses

The effects of interactions between inoculations and phosphorus (P) doses, as well as the effect of P_2_O_5_ doses, were not significant on leaf N concentrations of sugarcane; however, the effect of inoculations was significant on leaf N concentration (Table 1). The treatments with inoculation of PGPBs provided higher leaf N concentration in sugarcane compared to those without inoculation treatments. 

There was a significant interaction (*p* < 0.05) between inoculations with PGPBs and P_2_O_5_ doses for leaf P concentration of sugarcane (Table 1). 

Leaf P concentration of sugarcane plants was increased with inoculations of *Azospirillum brasilense*, *Pseudomonas fluorescens*, *Azospirillum brasilense* + *Bacillus subtilis*, *Azospirillum brasilense* + *Pseudomonas fluorescens* and *Bacillus subtilis* + *Pseudomonas fluorescens* in association with increasing residual P_2_O_5_ doses (Figure 1). Inoculations with *A. brasilense*, *P. fluorescens*, *A. brasilense* + *B. subtilis*, *A. brasilense + P. fluorescens*, *B. subtilis* + *P. fluorescens* and *A. brasilense* + *B. subtilis* + *P. fluorescens* at a residual P_2_O_5_ dose of 180 kg ha^−1^ were observed with higher leaf P concentration, which were statistically not different from the leaf P concentration obtained in the residual P_2_O_5_ dose of 135 kg ha^−1^ in combination with inoculations of *P. fluorescens*, *A. brasilense* + *B. subtilis*, *A. brasilense* + *P. fluorescens* and *B. subtilis* + *P. fluorescens* (Figure 1). 

### 2.2. Technological Attributes of Sugarcane as a Function of Inoculations and P_2_O_5_ Doses

Interactions between inoculations with PGPBs and phosphorus (P) doses were significant for % fiber, juice brix, pol, total recoverable sugar (ATR), stem yield (STY) and sugar yield (SUY) of cane sugar. The effect of the single treatment of inoculations and the residual dose of P_2_O_5_ and their interactions were not significant for the purity of the sugarcane juice (Table 2).

The highest fiber % was observed in the treatments with inoculations of *A. brasilense*, *P. fluorescens*, *A. brasilense* + *B. subtilis* and *A. brasilense* + *B. subtilis* + *P. fluorescens* without residual P fertilization. In addition, inoculations with *P. fluorescens*, *A. brasilense* + *P. fluorescens*, *B. subtilis* + *P. fluorescens* and *A. brasilense* + *B. subtilis* + *P. fluorescens* at a residual P_2_O_5_ dose of 45 kg ha^−1^ obtained the highest fiber % while the triple co-inoculation provided higher fiber % only at residual P_2_O_5_ dose of 90 kg ha^−1^. The treatments without inoculation and with co-inoculations of *A. brasilense* + *P. fluorescens* and *A. brasilense* + *Bacillus subtilis* + *Pseudomonas fluorescens* increased fiber % at a residual P_2_O_5_ dose of 135 kg ha^−1^, whereas inoculations with *A. brasilense*, *B. subtilis*, *P. fluorescens*, *A. brasilense* + *P. fluorescens* and *A. brasilense* + *B. subtilis* + *P. fluorescens* provided higher fiber % at a residual P_2_O_5_ dose of 180 kg ha^−1^ (Figure 2A).

Inoculation with *Azospirillum brasilense* + *Bacillus subtilis* reduced sugarcane fiber % with increasing residual P_2_O_5_ doses, indicating that increasing P fertilization doses collaborates effectively with the performance of these bacteria to reduce fiber %, which is undesirable for juice extraction. The inverse was observed with co-inoculation of *A. brasilense* + *P. fluorescens*, where fiber % of sugarcane was increased with increasing residual P doses, which is harmful to the quality of sugarcane (Figure 2A).

The percentage of apparent soluble solids (% brix) increased with increasing doses of residual P_2_O_5_ under co-inoculation of *B. subtilis* + *P. fluorescens* (Figure 2B). Inoculation with *B. subtilis* at the residual P_2_O_5_ dose of 45 kg ha^−1^ was observed with highest % of brix. The lowest % brix was determined in the treatments with inoculation of *P. fluorescens* and with co-inoculation of *A. brasilense* + *P. fluorescens* and *A. brasilense* + *B. subtilis* + *Pseudomonas fluorescens* at a residual P_2_O_5_ dose of 90 kg ha^−1^. Inoculations of *A. brasilense*, *B. subtilis*, *P. fluorescens* and *A. brasilense* + *P. fluorescens* associated with a residual P_2_O_5_ dose of 180 kg ha^−1^ provided lower % brix (Figure 2B).

There was an increase in pol percentage of sugarcane (% of pol) with increasing residual P_2_O_5_ doses up to 75 kg ha^−1^ in combination with inoculation of *Bacillus subtilis*, while a further increase in residual P_2_O_5_ doses led to the reduction in pol % in sugarcane. In addition, pol% of sugarcane was also reduced with increasing residual P_2_O_5_ doses under inoculations with *Pseudomonas fluorescens* and *A. brasilense* + *B. subtilis* (Figure 2C). 

### 2.3. Total Recoverable Sugar, Stalks and Sugar Yield as a Function of Inoculations and P Doses

The concentration of total recoverable sugar (TRS) was reduced in the treatments with inoculation of *P. fluorescens* and *A. brasilense* + *B. subtilis* + *P. fluorescens* as compared to other inoculations and co-inoculations at a residual P_2_O_5_ dose of 90 kg ha^−1^. The same effect was observed with inoculations of *A. brasilense*, *B. subtilis*, *P. fluorescens* and *A. brasilense + P. fluorescens* at a residual P_2_O_5_ dose of 180 kg ha^−1^. In addition, TRS concentration was increased with inoculation of *B. subtilis* under the increasing residual dose of P_2_O_5_ up to 72 kg ha^−1^ (Figure 3A). 

The treatments inoculated with *A. brasilense* and *B. subtilis* were observed with greater stalk productivity or stalk yield (STY) ha^−1^ and sugar yield (SUY) ha−^1^ in the absence of P_2_O_5_ fertilization in relation to other inoculations. Stalk yield and SUY were increased with inoculations of *A. brasilense*, *B. subtilis*, *P. fluorescens* and *A. brasilense* + *B. subtilis* under the residual P_2_O_5_ dose of 45 kg ha^−1^, while the same effect was observed with inoculation of *P. fluorescens* at the residual P_2_O_5_ dose of 90 kg ha^−1^, and with co-inoculation of *A. brasilense* + *P. fluorescens* at a P_2_O_5_ dose of 135 kg ha^−1^ (Figure 3A, C). The residual P_2_O_5_ dose of 180 kg ha^−1^ under inoculations of *A. brasilense*, *P. fluorescens*, *A. brasilense* + *B. subtilis* and *B. subtilis* + *P. fluorescens* were observed with greater STY as compared with other treatments at the same P_2_O_5_ dose. Sugarcane stalk yield was also increased in the treatments with inoculation of *P. fluorescens* and without inoculation in association with increasing residual P_2_O_5_ doses up to 102 and 114 kg ha^−1^, respectively. The treatments without inoculation and inoculation with *P. fluorescens* could produce a maximum STY of 110.5 and 142.8 t stalks ha^−1^ at calculated residual P_2_O_5_ dose of 102 and 114 kg ha^−1^. In addition, there was a reduction in STY in the treatments with inoculation of *B. subtilis* and increasing residual P_2_O_5_ doses (Figure 3B). 

Sugar yield (SUY) of sugarcane was increased without inoculation and with inoculation of *P. fluorescens* and *A. brasilense* + *P. fluorescens* under maximum residual P_2_O_5_ doses of 116, 98 and 110 kg ha^−1^, giving a maximum estimated SUY of 15, 16.7 and 16 t ha^−1^, respectively. Sugar yield was increased with increasing residual P_2_O_5_ doses under co-inoculation of *B. subtilis* + *P. fluorescens* and *A. brasilense* + *B. subtilis* + *P. fluorescens*. In addition, there was a reduction in SUY with increasing P doses and inoculations with *A. brasilense* and *B. subtilis* (Figure 3C). 

## 3. Discussion

Nitrogen and phosphorus are considered the most important nutrients for the growth and yield of crop plants. Leaf concentrations of N and P in the range of 18–26 and 1.5–3.0 g kg^−1^, respectively, are considered adequate for sugarcane ratoon leaves [22,23]. Leaf N concentrations in the present study are within the established standard range, except for treatments without inoculation (17.35 g kg^−1^) and with inoculation of *A. brasilense* (3.20 g ha^−1^). Leaf P concentration in treatments with inoculations of *A. brasilense* and *A. brasilense* + *B. subtilis* under residual P_2_O_5_ dose of 180 kg ha^−1^ was increased by 31 and 28% as compared to without inoculation. This may be due to the ability of PGPBs to improve plant growth directly and indirectly through various mechanisms [11], including nitrogen fixation and phosphate solubilization. This may be because of greater root growth (increasing soil exploitation by roots) and biological N fixation by *A. brasilense* [24] and increasing P availability and accumulation with inoculations of *Bacillus subtilis* and *P. fluorescens* [21]. The mechanism of P solubilization has been associated with the release of organic acids, which chelate cations of phosphate through hydroxyl and carboxyl groups and then convert them into soluble forms [25,26] with greater availability to the roots, thus increasing its uptake by plant roots and transport to the leaves, resulting in higher leaf P concentrations. It has also been reported that by-products enriched with phosphate rock and inoculated with P-solubilizing bacteria (*P. aeruginosa*, *Bacillus* sp. and *Rhizobium* sp.) in soils cultivated with sugarcane [27]. Rosa et al. [28] reported that inoculation and co-inoculation with *A. brasilense*, *B. subtilis* and *P. fluorescens* increased foliar P concentrations in the first sugarcane harvest. 

The % of sugarcane fiber was increased by 11.65% (Table 2; Figure 2A); this is within the agro-industrial quality of sugarcane within the standard range of 11–13% found in a previous study [29]. It is interesting and worth mentioning that % fiber values could not exceed the maximum value of 13%, as the higher % fiber content would negatively interfere at the time of juice extraction. Renan et al. [10] showed that the percentage of sugarcane fiber reflects on the juice extraction efficiency, where higher values of % fiber reduce the extraction efficiency. Furthermore, it should be considered that sugarcane varieties with low fiber content are more susceptible to mechanical damage during cutting and transport, thus impairing % brix, Pol and SUY.

Sugarcane requires low % brix, and minimum % brix values produced 18% higher sugar yield regardless of residual P doses and inoculations (Table 2). The % brix is directly related to sugar yield and the ideal % varies from 18 to 25% [30]. Inoculation with *B. subtilis* at a residual P_2_O_5_ dose of 45 kg ha^−1^ increased brix by 0.8% as compared to without inoculation with the highest value for the present work. This demonstrated the ability of these PGPBs in associate with residual P doses to improve quality of sugarcane juice (Figure 2B). It has previously reported that inoculation and co-inoculation with *A. brasilense*, *P. fluorescens* and *B. subtilis* under reduced P_2_O_5_ dose improved brix% in the first crop of sugarcane, proved that P solubilization favors the production of sugar [28].

The sugarcane pol% indicates all the apparent sucrose of the absolute juice of the harvested cane. Thus, increasing pol values could improve industrial yield of sugarcane [31]. The present study indicated that pol % of sugarcane was reduced in the treatments with increasing residual P_2_O_5_ doses under inoculation of *P. fluorescens* and co-inoculation of *A. brasilense* + *B. subtilis*, indicating that increase in P fertilization can reduce efficiency of these bacteria in the industrial sugarcane production. The co-inoculation with *A. brasilense* + *B. subtilis* + *P. fluorescens* provided the highest pol% of sugarcane in comparison to other inoculations, being at least 1.4% higher than the other inoculations (Figure 2C).

Inoculation with *B. subtilis* and *A. brasilense* + *B. subtilis* at the residual P_2_O_5_ dose of 45 kg ha^−1^ provided the highest total recoverable sugar (TRS) in relation to the other P doses within the same inoculation (Figure 3A), which allows us to infer that these bacteria benefited from the low P_2_O_5_ dose. Phosphate fertilizer allowing the release of compounds that could help to improve industrial quality of sugarcane juice of this variety. Rosa et al. [28] describes that inoculation and co-inoculation with *A. brasilense*, *B. subtilis* and *P. fluorescens* provided higher TRS of sugarcane juice under reduced P doses. 

At a residual P_2_O_5_ dose of 45 kg ha^−1^, the stalk yield (STY) (t ha^−1^) was increased with inoculation of *A. brasilense*, *B. subtilis*, *P. fluorescens* and *A. brasilense* + *B. subtilis* + *P. fluorescens*, being 28, 21, 18 and 16% higher in relation to without inoculation, respectively (Figure 3B). Inoculation with Pseudomonas fluorescens increased the STY of sugarcane by 17% (17.96 t ha^−1^) as compared to without inoculation at a residual P_2_O_5_ dose of 90 kg ha^−1^. In addition, co-inoculation with *A. brasilense* + *P. fluorescens* at residual P_2_O_5_ dose of 135 kg ha^−1^ provided 42% (36.69 t ha^−1^) higher STY than without inoculation, being the highest STY observed in the entire study, also allowing a 25% reduction in phosphate fertilization. It was reported in a previous study that ratoon cane productivity was significantly improved with joint application of PGPBs, green manure and adequate fertilizers [27]. 

The P use efficiency for the treatments with inoculation of *A. brasilense*, *B. subtilis* and *P. fluorescens* at a residual P_2_O_5_ dose of 45 kg ha^−1^, were 0.76, 0.62 and 0.58 t of sugarcane kg^−1^ P_2_O_5_, respectively. For inoculation with *Pseudomonas fluorescens* at a residual P_2_O_5_ dose of 90 kg ha^−1^, the P use efficiency was 0.54 t of sugarcane kg^−1^ P_2_O_5_. The P use efficiency for the co-inoculation with *A. brasilense* + *P. fluorescens* at residual P_2_O_5_ dose of 135 kg ha^−1^ was 0.38 t of sugarcane kg^−1^ P_2_O_5_. 

Sugar yield (SUY) (t ha^−1^) was increased with inoculation of *A. brasilense*, *B. subtilis*, *P. fluorescens* and *A. brasilense* + *B. subtilis* + *P. fluorescens* at a residual P_2_O_5_ dose of 45 kg ha^−1^, being 26, 23, 14 and 17% higher as compared to without inoculation treatments (Figure 3C). Inoculation with *P. fluorescens* and co-inoculation with *A. brasilense + P. fluorescens* at a residual P_2_O_5_ dose of 90 kg ha^−1^ provided 8 and 43% higher SUY, respectively, in comparison of without inoculated treatments (Table 2; Figure 3C). The previous study with the first sugarcane crop indicated that co-inoculation of *A. brasilense + B. subtilis* reduced P_2_O_5_ fertilization by 75% during first sugarcane crop while inoculation with *A. brasilense* improved agro-industrial quality of sugarcane under P_2_O_5_ dose of 45 kg ha^−1^ [28]. 

The exact mechanisms of inoculation with *A. brasilense* and *B. subtilis* on the productivity and industrial quality of sugarcane is still needed to be disclosed. However, use of these microorganisms increased availability and absorption of P, which is mainly responsible for high leaf P concentration, growth, industrial quality and yield of sugarcane crops [32,33,34]. The gene sequencing of *A. brasilense* (Ab-V5 and Ab-V6 strains) highlighted its role in auxin synthesis [34], nutrient availability and cycling [14,35] and biological N fixation [36,37]. Furthermore, *B. subtilis* has been described as having the potential ability to promote plant growth, solubilize P, and inhibit infestation of phyto-pathogenic attack and heavy metal accumulation [38,39]. *Bacillus* sp. is a genus that is characterized by being an excellent phosphate solubilizer, including *B. subtilis*, and P is considered one of the most critical nutrients to increase crop yield [40,41]. Studies have described that plants adapt different mechanisms under inoculation with *A. brasilense* and *B. subtilis* to potentially increase soil P availability and food security [39,42]. Bacteria of the genera *Arthrobacter*, *Bacillus*, *Pseudomonas* and *Rhizobium*, among others, have demonstrated the ability to solubilize phosphates [17]. Therefore, these PGPBs have the potential to make better use of phosphorus present in the solid phase of the soil, constituting viable alternatives for the inoculation of crops. However, it is noteworthy that the genera *Bacillus*, *Pseudomonas* and *Rhizobium* are considered the most efficient solubilizers of inorganic phosphates [15]. 

Single inoculation with *P. fluorescens* and/or co-inoculation with *A. brasilense* and *B. subtilis* increased leaf P concentration, industrial quality and productivity of sugarcane in the current study. It may be possible due to the efficiency of *P. fluorescens* bacterium to use metabolites as a bio-control agent with the synthesis of antibiotics and volatile organic compounds to combat soil pathogens [43,44]. This is promising in terms of phosphate solubilization [45] and N-fixing activities [46]. The genus *Pseudomonas* sp. produces gluconic acid, which is among the different types of organic acids involved in the solubilization of phosphate, which is considered as an important technique to improve management of P fertilization in modern agriculture [47]. Therefore, co-inoculation with *A. brasilense* + *P. fluorescens* can be an important tool in the sustainable management of phosphate fertilization in sugarcane cultivated in tropical soils, optimizing the use of residual P_2_O_5_ doses and allowing to reduce P_2_O_5_ fertilization by 25% in the ratoon cane field.

## 4. Materials and Methods

### 4.1. The Location of the Experimental Area

This experiment was conducted during the 2019/20 cropping season at Limoeiro Farm, a part of the ethanol and sugar industry in Ilha Solteira, in the northwest of the state of São Paulo, Brazil. The experimental area is located at the geographical coordinates of 20°21′14″ S latitude and 51°04′51″ W longitude, with an altitude of 371 m. The experiment was conducted during the third year of sugar-cane cultivation (second ratoon). The climate in the region is Aw (Köppen scale, defined as tropical with dry winters). Climatic data referring to the period of the experiment were recorded (Figure 4).

The cultivation history of the area in the last ten years and prior to the implantation of sugarcane (2017) showed that this area was cultivated with a pasture (*Urochloa brizantha*) with some degree of degradation. The soil was classified as Rhodic Haplustox according to USDA [48], and as medium-to-sandy-textured Red Dystrophic soil according to SiBCS [49], with a granulometry of 777, 98, 125 g kg^−1^ and 747, 88, 165 g kg^−1^ of sand, silt and clay at depths of 0.00–0.25 and 0.25–0.50 m, respectively. The chemical attributes of the soil were determined before implantation of sugarcane and regrowth of second ratoon cane crop (Table 3). 

### 4.2. Experimental Design and Treatments

The experiment was designed in randomized blocks with three replications and arranged in an 8 × 5 factorial scheme. The treatments consisted of eight inoculations (1. No inoculation (control); 2. Inoculation with *Azospirillum brasilense*; 3. Inoculation with *Bacillus subtilis*; 4. Inoculation with *Pseudomonas fluorescens*; 5. Inoculation with *Azospirillum brasilense* + *B. subtilis*; 6. Inoculation with *Azospirillum brasilense* + *Pseudomonas fluorescens*; 7. Inoculation with *Bacillus subtilis* + *Pseudomonas fluorescens*; 8. Inoculation with *Azospirillum brasilense* + *Bacillus subtilis* + *Pseudomonas fluorescens*) and five residual doses of P_2_O_5_ (0, 45, 90, 135 and 180 kg ha^−1^). Phosphate fertilizer doses were applied at the time of sugarcane plantation in 2017 from the source of triple superphosphate 46% P_2_O_5_, corresponding to 0, 25, 50, 75 and 100% of the recommended P_2_O_5_ dose according to [50]. These PGPBs are commercially used in Brazil with strains of *A. brasilense* (AzoTotal™), *B. subtilis* (Vult™) and *P. fluorescens* (Audax™). The plots consisted of 5 lines 5 m in length and spaced at 1.5 m, considering the three central lines of each plot for evaluations. 

### 4.3. Installation and Conduct of the Experiment

The experimental field was applied with soil 1.0 t ha^−1^ of dolomitic limestone (PRNT 85%) to obtain 60% base saturation as well as 1.0 t ha^−1^ of gypsum to raise sulfur content (S) after initial soil chemical characterization and 15 days before planting, as recommended for sugarcane cultivation. Three harrowing and one sub-soiling operations were carried out, followed by furrowing at a depth of 0.40 m and applying the insecticide fipronil (180 g ha^−1^ of ai) + fungicide pyraclostrobin (125 g ha^−1^ of ai) in planting furrow. 

Manual plantation of a sugarcane variety RB92579 was carried out in planting furrows on 11 July 2017, containing about 22 buds m^−1^. All the treatments were equally fertilized with 30 kg ha^−1^ of N (ammonium nitrate, 33% N) and 120 kg ha^−1^ of K_2_O (potassium chloride, 60% K_2_O) based on soil analysis. In addition, respective doses of P_2_O_5_ were applied in a single dose for each treatment in the planting furrow at the time of sugarcane implantation in 2017.

Fertilization of topdressing in the second ratoon was performed 30 days after sprouting (DAS) on 12 September 2019 from ammonium nitrate as a source of N and potassium chloride as a source of K_2_O based on the recommendations of [23], as well as land slope and precipitation. The inoculation with PGPBs was performed at 30 DAS of the second ratoon sugarcane crop, using the following liquid inoculants doses: 1.0 L ha^−1^ of *A. brasilense* (strains Ab-V5 and Ab-V6 (guarantee of 2 × 10^8^ colony forming units (CFU) mL^−1^)), 0.5 L ha^−1^ of *B. subtilis* (strains CCTB04 (guarantee of 1 × 10^8^ CFU mL^−1^)) and 0.5 L ha^−1^ of *P. fluorescens* (strains CCTB03 (guarantee of 2 × 10^8^ CFU mL^−1^)) on the basis of manufacturer’s recommendations. The spray volume was 200 L ha^−1^ (for all applied inoculants), applied with a backpack sprayer at the base of the second ratoon cane tillers during early morning due to milder temperature.

The correction of boron and zinc deficiency was carried out by applying 5 kg ha^−1^ of zinc (zinc sulfate, 20% Zn and 10% S) and 2 kg ha^−1^ of boron (boric acid, 17% B) in coverage at about 15 cm from the base of the plants at 60 DAS.

Weeds, pests and diseases were controlled via chemicals and biological ways according to the crop needs throughout the growth cycle. The Insecticides Chlorantraniliprole (10 g ha^−1^ ai) + Lambda cyhalothrin (5 g ha^−1^ ai) were applied. Four releases of *Trichogramma galloi* were performed throughout the crop cycle at 142, 148, 213 and 220 DAS to biologically control sugarcane borer (*Diatraea saccharalis*). The ratoon sugarcane was manually harvested on the 5th August 2020, at 357 DAS. 

### 4.4. Assessments

The middle third of 12 flag leaves per plot were collected at the full vegetative growth phase (157 DAS) in the morning, and the midrib was removed as described by [50] for the determination of leaf N and P concentrations [51]. The weight was quantified at the time of harvest in order to estimate stalk yield per hectare (STY) (t ha^−1^).

Ten commercial-quality sugarcane stalks were collected from each plot at the time of harvest and determined for technological quality at Technological Analysis Laboratory by following the methodology defined in the System Payment of Sugarcane Based on Sucrose Content (SPSBSC), according to [52]. Fiber (%), juice purity (%), soluble solids or juice brix content (brix %), apparent sucrose content of cane (Pol-%) and total recoverable sugar (TRS) (kg sugar per t cane^−1^) were also evaluated at the technological laboratory of ethanol and sugar industry at Suzanápolis, Brazil. Sugar yield SUY (t sugar ha^−1^) was quantified from the product of STY and pol %, divided by 100 in each individual plot. 

### 4.5. Statistical Analysis

The results were submitted to analysis of variance (F test) and Scott–Knott tests (*p* ≤ 0.05) to compare inoculation means, and regression equations were adjusted for the residual effect of P_2_O_5_ doses. Statistical analyses were processed using the computer program SISVAR [53], and graphs were plotted with the aid of SigmaPlot 12.5 software.

## 5. Conclusions

Inoculation with *Pseudomonas fluorescens* alone, and the co-inoculations *Azospirillum brasilense* + *B. subtilis*, *Azospirillum brasilense* + *Pseudomonas fluorescens* and *Bacillus subtilis* + *Pseudomonas fluorescens* under residual P_2_O_5_ doses of 135 and 180 kg ha^−1^ increased leaf P concentration. In addition, the isolated or combined inoculation with *Azospirullum brasilense*, *Pseudomonas fluorescens* and *Bacillus subtilis* promoted leaf N concentration in relation to control.

The co-inoculation of *Azospirillum brasilense* + *Pseudomonas fluorescens* in association with a residual P_2_O_5_ dose of 45 kg ha^−1^ improved agro-industrial quality of the second ratoon cane. Co-inoculation with *Azospirillum brasilense* + *Pseudomonas fluorescens* at a residual P_2_O_5_ dose of 135 kg ha^−1^ increased stalk yield by 42% and sugar by 43%, providing a 25% reduction in phosphate fertilization in the second ratoon cane.

## Figures and Tables

**Figure 1 plants-12-02699-f001:**
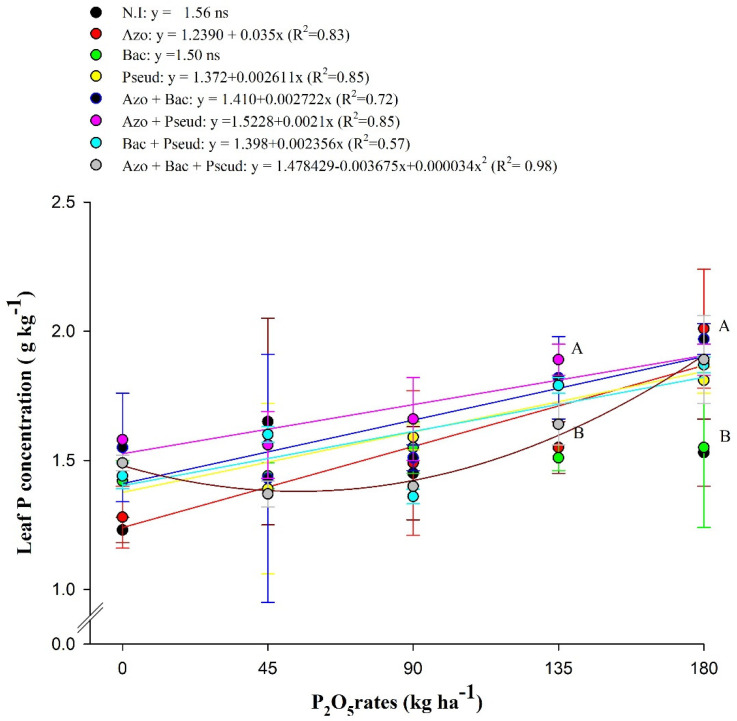
Interaction of residual P_2_O_5_ doses within inoculations for leaf phosphorus (P) concentration of the second ratoon sugarcane crop (Mean values ± standard deviation). N.I (non-inoculation), Azo (*Azospirillum brasilense*); Bac (*Bacillus subtilis*); Pseud (*Pseudomonas fluorescens*); Azo + Bac (*Azospirillum brasilense* + *B. subtilis*); Azo + Pseud (*Azospirillum brasilense + Pseudomonas fluorescens*); Bac + Pseud *(Bacillus subtilis + Pseudomonas fluorescens*); Azo + Bac + Pseud (*Azospirillum brasilense + Bacillus subtilis + Pseudomonas fluorescens*). Error bars indicate standard deviation of the means (*n* = 4 replications). Different letters in each P_2_O_5_ dose are statistically different according to the Scott–Knott test at 5% probability.

**Figure 2 plants-12-02699-f002:**
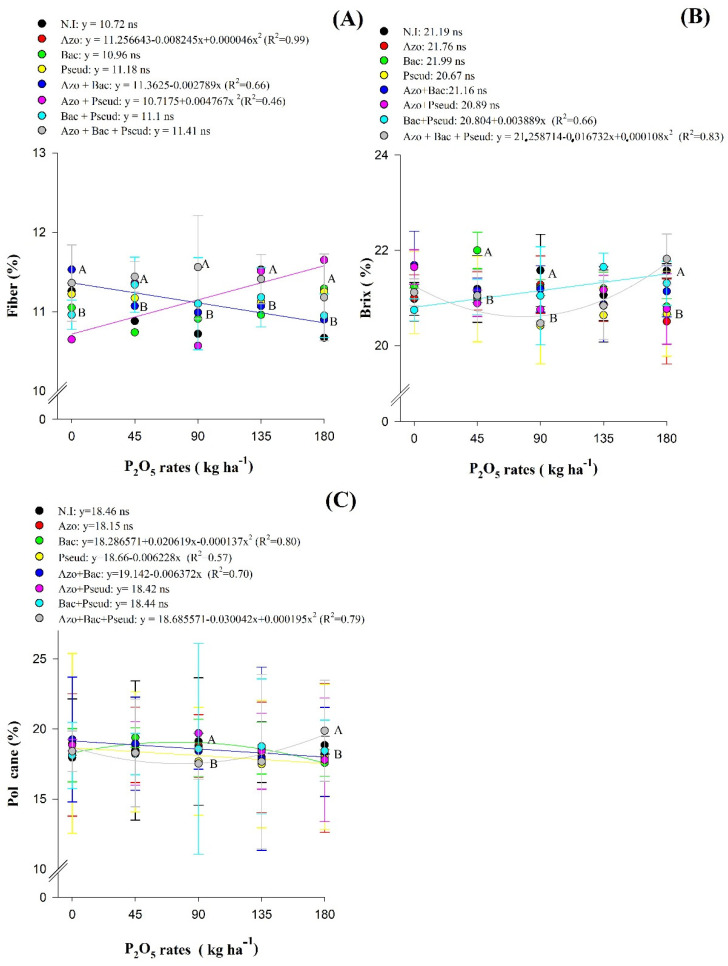
Interaction of residual P_2_O_5_ doses within inoculations for Fiber (**A**), brix (**B**) and Pol (**C**) of the second ratoon variety RB92579 (Mean values ± standard deviation). NI (non-inoculation); Azo (*Azospirillum brasilense*); Bac (*Bacillus subtilis*); Pseud (*Pseudomonas fluorescens*); Azo + Bac (*Azospirillum brasilense + B. subtilis*); Azo + Pseud (*Azospirillum brasilense + Pseudomonas fluorescens*); Bac + Pseud (*Bacillus subtilis + Pseudomonas fluorescens*); Azo + Bac + Pseud (*Azospirillum brasilense + Bacillus subtilis + Pseudomonas fluorescens*). Error bars indicate standard deviation of the means (*n* = 4 replications). Different letter in each P_2_O_5_ dose were statistically different according to the Scott–Knott test at 5% probability.

**Figure 3 plants-12-02699-f003:**
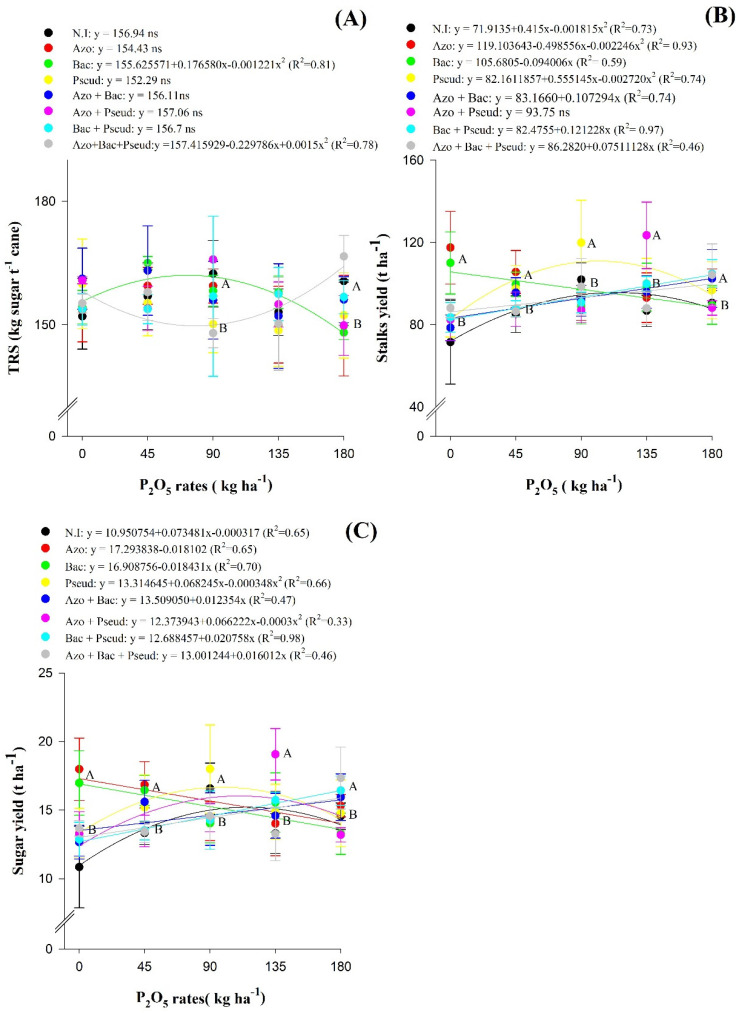
Interaction of residual P_2_O_5_ doses within inoculations for total recoverable sugar—TRS (**A**), stalk yield—STY (**B**) and sugar yield—SUY (**C**) of the second ratoon cane variety RB92579 (Mean values ± standard deviation). NI (non-inoculation); Azo (*Azospirillum brasilense*); Bac (*Bacillus subtilis*); Pseud (*Pseudomonas fluorescens*); Azo + Bac (*Azospirillum brasilense + B. subtilis*); Azo + Pseud (*Azospirillum brasilense + Pseudomonas fluorescens*); Bac + Pseud (*Bacillus subtilis + Pseudomonas fluorescens*); Azo + Bac + Pseud (*A. brasilense + Bacillus subtilis + Pseudomonas fluorescens*). Error bars indicate standard deviation of the means (*n* = 4 replications). Different letters in each P_2_O_5_ dose are statistically different according to the Scott–Knott test at 5% probability.

**Figure 4 plants-12-02699-f004:**
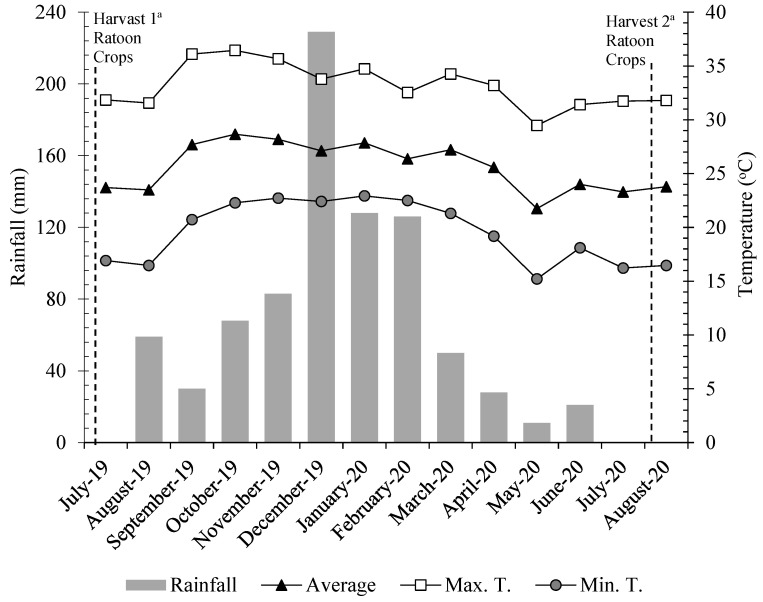
Monthly average precipitation, maximum, average and minimum temperatures during the period of experiment. This data was collected from the automated weather station at experimental location. Ilha Solteira, SP, 2019/2020.

**Table 1 plants-12-02699-t001:** Leaf nitrogen (N) and phosphorus (P) concentrations of second ratoon cane of the RB92579 variety as a function of inoculations and P doses.

P_2_O_5_ Doses (Kg ha^−1^)	Leaf N Concentration	Leaf P Concentration
	g kg^−1^	
0	19.42	1.43
45	19.23	1.48
90	19.97	1.50
135	19.33	1.69
180	20.15	1.82
**Inoculations**		
Without inoculation	17.35 b	1.48
*Azospirillum brasilense*	20.03 a	1.55
*Bacillus subtilis*	19.97 a	1.49
*Pseudomonas fluorescens*	19.78 a	1.61
*Azospirillum brasilense* + *Bacillus subtilis*	20.20 a	1.65
*Azospirillum brasilense* + *Pseudomonas fluorescens*	19.73 a	1.72
*Bacillus subtilis* + *Pseudomonas fluorescens*	20.12 a	1.61
*Azospirillum brasilense* + *Bacillus subtilis* + *Pseudomonas fluorescens*	19.76 a	1.56
**Test F**		
P_2_O_5_ doses (D)	ns	**
Inoculation (I)	*	*
D × I	ns	*
Overall Means	19.62	1.58
standard Error	0.62	0.05
**CV (%)**	9.93	9.84

Means followed by the same letter in the column were statistically not different according to the Scott–Knott test at 5% probability. TRS: total recoverable sugar, STY: stalk yield and SUY: sugar yield. CV: coefficient of variance. **, * and ns: significant at 1% and 5% at *p* < 0.01, *p* < 0.05 and non-significant, respectively. *n* = 4 replications.

**Table 2 plants-12-02699-t002:** Indicators of technological quality, stalk yield (STY) and sugar yield (SUY) of the second ratoon cane of the RB92579 variety as a function of P dose and inoculations.

P_2_O_5_ Doses (Kg ha^−1^)	Fiber	Purity	Brix	Pol	TRS	STY	SUY
	%	%	%	%	kg Sugar t^−1^ Cane	t ha^−1^	t ha^−1^
0	11.16	87.27	21.19	18.47	156.33	89.37	13.94
45	11.12	87.53	21.18	18.59	157.80	93.19	14.72
90	10.98	87.84	20.99	18.56	157.10	95.91	15.05
135	11.22	86.93	21.04	18.14	153.03	98.19	15.04
180	11.14	86.93	21.08	18.35	154.99	97.39	17.10
**Inoculation**							
Without inoculation	11.01	87.74 a	21.27	18.54	157.00	87.21	13.72
*Azospirillum brasilense*	11.07	87.72 a	20.96	18.35	154.44	101.52	15.66
*Bacillus subtilis*	10.99	86.56 a	21.29	18.47	156.69	97.22	15.25
*Pseudomonas fluorescens*	11.17	87.41 a	20.77	18.09	153.23	99.53	15.23
*Azospirillum brasilense* + *Bacillus subtilis*	11.11	87.38 a	21.21	18.57	157.70	92.82	14.62
*Azospirillum brasilense* + *Pseudomonas fluorescens*	11.15	86.84 a	21.05	18.60	157.07	93.75	14.69
*Bacillus subtilis* + *Pseudomonas fluorescens*	11.11	87.49 a	21.15	18.42	155.72	93.39	14.56
*Azospirillum brasilense* + *Bacillus subtilis* + *Pseudomonas fluorescens*	11.39	87.25 a	21.06	18.36	154.96	93.04	14.44
**Test F**							
P_2_O_5_ doses (D)	ns	ns	ns	ns	ns	**	*
Inoculation (I)	*	ns	*	ns	ns	**	*
D × I	**	ns	**	**	**	**	**
Overall Means	11.12	87.29	09.21	18.42	155.85	94.81	14.77
Standard Error	0.078	0.41	0.12	0.20	1.66	2.29	0.37
**CV (%)**	3.12	2.12	2.55	4.87	4.77	10.83	11.34

Means followed by the same letter in the column were statistically not different according to the Scott–Knott test at 5% probability. TRS: total recoverable sugar, STY: stalk yield and SUY: sugar yield. CV: coefficient of variance. **, * and ns: significant at 1% and 5% at *p* < 0.01, *p* < 0.05 and non-significant, respectively. *n* = 4 replications.

**Table 3 plants-12-02699-t003:** Chemical characterization of the soil in the experimental area before planting sugarcane and regrowth of the second ratoon cane in the 0.00–0.25 and 0.25–0.50 m layers.

Before Planting Sugarcane										
Layers (m)	P	S-SO_4_	OM	pH	K	Ca	Mg	H + Al	Al	Sb
	mg dm^−3^		g dm^−3^	CaCl_2_	mmol_c_ dm^−3^					
0.00–0.25	2	3	13	4.7	2.6	8	6	20	1	16.6
0.25–0.50	2	2	12	4.8	2.4	9	7	20	2	18.4
**Layers** **(m)**	**B ^a^**	**Cu ^b^**	**Fe ^b^**	**Mn ^b^**	**Zn^b^**	**CEC**	**V**	**m**		
	mg dm^−3^					mmol_c_ dm^−3^	%	%		
0.00–0.25	0.22	0.8	14	16.2	0.6	36.6	45	6		
0.25–0.50	0.22	1.0	7	8.3	0.3	38.4	48	10		
**Before regrowth of 2nd ratoon cane**										
**Layers** **(m)**	**P**	**S-SO_4_**	**OM**	**pH**	**K**	**Ca**	**Mg**	**H + Al**	**Al**	**Sb**
	mg dm^−3^		g dm^−3^	CaCl_2_	mmol_c_ dm^−3^					
0.00–0.25	7	2	16	5	1	9	7	23	2	18
0.25–0.50	5	2	11	5	1	9	6	20	2	17
**Layers** **(m)**	**B ^a^**	**Cu ^b^**	**Fe ^b^**	**Mn ^b^**	**Zn ^b^**	**CEC**	**V**	**M**		
	mg dm^−3^					mmol_c_ dm^−3^	%	%		
0.00–0.25	0	1	25	21	1	41	44	10		
0.25–0.50	0	1	16	12	1	37	45	13		

^a^ Determined in hot water, ^b^ Determined in DTPA (diethylenetriaminepentaacetc acid). OM: organic matter. CEC: cation exchange capacity, SB: sum of bases, V: base saturation, M: Al saturation.

## Data Availability

The original contributions presented in the study are included in the article, further inquiries can be directed to the corresponding author.

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
