# Peer review of "Technological Quality of Sugarcane Inoculated with Plant-Growth-Promoting Bacteria and Residual Effect of Phosphorus Rates"

_plants, 2023, doi:10.3390/plants12142699_

Round 1
Reviewer 1 Report
This manuscript (plants-2503546) investigated the impact of phosphate fertilisation and plant growth-promoting bacteria (PGPBs) on 2nd ratoon cane crop's nutrient levels and productivity. The results showed that inoculations, especially with Pseudomonas fluorescens and in combination, significantly reduce phosphate fertilisation needs and boost productivity.
The topics are well-written. The introduction, results, discussion, materials and methods sections are well-informed and informative. The concept of the work and statistics are good. Some figures are of low quality, perhaps due to a conversion process to PDF? Please check and improve the quality in SigmaPlot. Minor corrections in English grammar and spelling are needed.
Minor comments:
The abstract requires an introductory sentence before stating the aim of the study. Additionally, consider reformulating to make it simpler and more direct.
Keywords in alphabetic order;
In the first instance, the abbreviations for Iron (Fe) and Aluminum (Al) should be clearly defined (L55 and L77).
Table 1, 2, what is "n" of sample? Change for F test;
Standardising the notation as 'P<0.05' and 'P≥0.05'.
What type of statistics are represented in Figure 1, 2?
Table 2 in material and methods? Check!
L423. space [51];
Minor corrections in English grammar and spelling are needed.
Author Response
Reviewer 1:
This manuscript (plants-2503546) investigated the impact of phosphate fertilisation and plant growth-promoting bacteria (PGPBs) on 2nd ratoon cane crop's nutrient levels and productivity. The results showed that inoculations, especially with Pseudomonas fluorescens and in combination, significantly reduce phosphate fertilisation needs and boost productivity.
The topics are well-written. The introduction, results, discussion, materials and methods sections are well-informed and informative. The concept of the work and statistics are good. Some figures are of low quality, perhaps due to a conversion process to PDF? Please check and improve the quality in SigmaPlot. Minor corrections in English grammar and spelling are needed.
R: Our most sincere gratitude to the reviewer, who took time from his busy schedule to help us making this manuscript a better paper. We hope that we have answered every inquiry to your satisfaction and also hope that you will find this version of publishable quality. We hope that this version has met the expectations of the reviewer.
The resolution of the figure is extended to 300 DPI.
We carefully proofread the entire manuscript and correct errors in English grammar and spelling.
Thanks!
Minor comments:
The abstract requires an introductory sentence before stating the aim of the study. Additionally, consider reformulating to make it simpler and more direct.
R: We added an introductory sentence before stating the aim of the study. Thanks!
Keywords in alphabetic order;
R: We got it right.
In the first instance, the abbreviations for Iron (Fe) and Aluminum (Al) should be clearly defined (L55 and L77).
R: We agree with reviewer and corrected it.
Table 1, 2, what is "n" of sample? Change for F test;
Standardising the notation as 'P<0.05' and 'P≥0.05'.
R: Added the number of samples, and we standardize it. Thanks!
What type of statistics are represented in Figure 1, 2?
R: Different letter in each P2O5 dose were statistically different by Scott-Knott test at 5% probability.
Table 2 in material and methods? Check!
L423. space [51];
R: Checked and corrected.
Comments on the Quality of English Language
Minor corrections in English grammar and spelling are needed.
R: We carefully proofread the entire manuscript and correct errors in English grammar and spelling.
Thanks!
Reviewer 2 Report
The article tried to evaluate the effect of inoculation with three species of plant growth-promoting bacteria, applied alone or in combination in association with reduced P2O5 doses (applied at the time of planting) on leaf N and P con-centrations, stalk yield, sugar and technological quality of the 2nd ratoon cane in tropical soil with low P content. The article is well written but Nitrogen results are mostly non significant, however are valuable to the readers. some parts of discussion required improvement as it looks repeated with results for example line 249-266. The discussion ipresented in lines 282-307 is a simple repetition with results with novalid reasons of results and probabilities that why these results are important in the context.
minor english checks required
Author Response
Reviewer 2:
The article tried to evaluate the effect of inoculation with three species of plant growth-promoting bacteria, applied alone or in combination in association with reduced P2O5 doses (applied at the time of planting) on leaf N and P con-centrations, stalk yield, sugar and technological quality of the 2nd ratoon cane in tropical soil with low P content. The article is well written but Nitrogen results are mostly non significant, however are valuable to the readers. some parts of discussion required improvement as it looks repeated with results for example line 249-266. The discussion ipresented in lines 282-307 is a simple repetition with results with novalid reasons of results and probabilities that why these results are important in the context.
R: Our most sincere gratitude to the reviewer, who took time from his busy schedule to help us making this manuscript a better paper. We hope that we have answered every inquiry to your satisfaction and also hope that you will find this version of publishable quality.
We improved the suggested parts in the manuscript. Also, we carefully proofread the entire manuscript and correct errors in English grammar and spelling. We hope that this version has met the expectations of the reviewer.
Thanks!